# Coupling Process-Based Crop Model and Extreme Climate Indicators with Machine Learning Can Improve the Predictions and Reduce Uncertainties of Global Soybean Yields

Qing Sun [1], Yi Zhang [1], Xianghong Che [2,*], Sining Chen [3], Qing Ying [4], Xiaohui Zheng [5] and Aixia Feng [6]

[1] State Key Laboratory of Severe Weather (LASW), Chinese Academy of Meteorological Sciences, Beijing 100081, China
[2] Chinese Academy of Surveying & Mapping, Beijing 100830, China
[3] Tianjin Climate Center, Tianjin 300074, China
[4] Earth System Science Interdisciplinary Center (ESSIC), University of Maryland, College Park, MD 20737, USA
[5] China Meteorological Administration Training Center, Beijing 100081, China
[6] National Meteorological Information Center, Beijing 100081, China
[*] Correspondence: chexh@casm.ac.cn

**Abstract:** Soybean is one of the most important agricultural commodities in the world, thus making it important for global food security. However, widely used process-based crop models, such as the GIS-based Environmental Policy Integrated Climate (GEPIC) model, tend to underestimate the impacts of extreme climate events on soybean, which brings large uncertainties. This study proposed an approach of hybrid models to constrain such uncertainties by coupling the GEPIC model and extreme climate indicators using machine learning. Subsequently, the key extreme climate indicators for the globe and main soybean producing countries are explored, and future soybean yield changes and variability are analyzed using the proposed hybrid model. The results show the coupled GEPIC and Random Forest (GEPIC+RF) model (R: 0.812, RMSD: 0.716 t/ha and rRMSD: 36.62%) significantly eliminated uncertainties and underestimation of climate extremes from the GEPIC model (R: 0.138, RMSD: 1.401 t/ha and rRMSD: 71.57%) compared to the other five hybrid models (R: 0.365–0.612, RMSD: 0.928–1.021 and rRMSD: 47.48–52.24%) during the historical period. For global soybean yield and those in Brazil and Argentina, low-temperature-related indices are the main restriction factors, whereas drought is the constraining factor in the USA and China, and combined drought–heat disaster in India. The GEPIC model would overestimate soybean yields by 13.40–27.23%. The GEPIC+RF model reduced uncertainty by 28.45–41.83% for the period of 2040–2099. Our results imply that extreme climate events will possibly cause more losses in soybean in the future than we have expected, which would help policymakers prepare for future agriculture risk and food security under climate change.

**Keywords:** soybean yield; extreme climate events; machine learning; crop model; uncertainty

## 1. Introduction

Soybean is the fifth most grown crop in the world [1]. Extreme weather and climate events including extreme temperature and precipitation [2], drought [3] and combined heat–drought disasters [4,5] will greatly impact the stability of global soybean production. The extreme heat stress would reduce soybean yield by a quarter by the 2080s [6]. The variations of temperature and precipitation may suppress 30% of US soybean yield [7]. The compound hot–dry extreme weather may reduce more than 0.8 t/ha of the soybean yield in the US [4]. And the soybean yield may be reduced by climate change about 1% in China [8]. Climate change and the increasing frequency of extreme weather and climate events not only lead to a decline in soybean yield and farmers' stable incomes, but also aggravate disaster risks, the demand for additional arable land, and the unstable supply of

food, which are seriously affecting global food security [9–11]. Furthermore, most of the main soybean producing regions are rainfed, which will increase the uncertainty and risk for soybean production and supply. Therefore, understanding how soybean yields change under extreme weather events in the future and its uncertainties become increasingly important [12–15].

The statistical models [4,16] and process-based crop models [17,18] are the two main approaches for predicting yield and quantitatively assessing the impact of meteorological disasters on yield. Different models have distinct applicability and simulation capabilities, making it necessary to identify the strengths and weaknesses of each model for better simulating accurate soybean yield prediction [19,20]. However, most crop models have large uncertainties in simulating crop yields under the influence of extreme disasters [12,13]. Soybean crop models are currently subject to large uncertainties and sometimes even have opposite trends in future projections [1]. This may be due to the lack of in-depth understanding of the mechanisms of extreme weather affecting crops, and the lack of relevant computational modules [21], or the modules are relatively simple and insufficient to accurately simulate the complex mechanisms of disaster impact [22,23].

In recent years, coupling crop models or statistical models with machine learning or deep learning methods has been obtaining recognition. Feng et al. [24] and Everingham et al. [25] incorporated the APSIM model and random forest (RF) model to increase the simulation performance for crop yield. Nearly 90% accuracy was achieved on sugarcane yield simulations when combining the crop model and multi-linear regression [26]. The DSSAT model can be incorporated with the Support Vector Regression (SVM) model to assess groundwater variability [27]. Moreover, the uncertainties also can be reduced using machine learning algorithms to constrain the simulated yield from crop model [15].

In this study, we aim to combine the process-based crop model and machine learning methods to build a hybrid model and constrain uncertainties from the crop model for global soybean yield projections. Besides mean climate state, we focus on assessing the impacts of extreme climate events during soybean growing season, which is the main reason why annual yield fluctuates and tends to be underestimated in previous studies [13,28]. The main objectives are to (1) analyze which type of machine learning methods are best for the hybrid model, (2) quantify the relative importance of extreme climate events during soybean growing season for the whole globe and for the main producing countries, (3) explore how the effects of the main extreme climate events will change for soybean in the future, and (4) compare the yield differences between the crop model and the hybrid model under future climate change at global and country scales.

## 2. Materials and Method

### 2.1. Process-Based Crop Model and Input Climate Data

A process-based crop model, the GIS-based Environmental Policy Integrated Climate model (GEPIC) [29], is adopted in this study as an example within the Inter-Sectoral Impact Model Intercomparison Project Phase 2b (ISIMIP2b) [30]. The GEPIC model was based on the Environmental Policy Integrated Climate model (EPIC) which could simulate crop yield and water productivity and assess the cost of erosion for determining optimal management strategies on a global scale [31].

Five bias-corrected climatic datasets, HadGEM2-ES, IPSL-CM5A-LR, MIROC-ESM-CHEM, GFDL-ESM2M and NorESM1-M [32–34], are the climate inputs of the GEPIC model. These datasets are from the General Circulation Models (GCMs) of the CMIP5 archives with history (1981–2005) and future (2040–2099) under Representative Concentration Pathway 6.0 (RCP 6.0). The grid resolutions of climatic datasets were 0.5° × 0.5° and we utilized daily maximum temperature, minimum temperature, mean temperature, precipitation and solar radiation. Other detailed inputs and simulation protocol such as crop calendar, soil properties and crop management can be found in the literature [17,30,35,36]. The outputs from the GEPIC model included simulated soybean yields, growing seasons (planting date

and maturity date) and the soybean evapotranspiration. Both crop model outputs and climatic datasets were downloaded from https://data.isimip.org (accessed on 18 July 2020).

### 2.2. Extreme Climate Indicators

Only when extreme climate occurs during soybean growing season does it affect soybean yield. In this study, we intended to assess the impacts of five types of extreme climate disasters, cold, heat, drought, waterlogging and compound heat–drought, during soybean main growing season (between planting and harvest days) that are relevant to soybean yield [14,24,37]. The soybean growing seasons were from the GEPIC model and were based on the SAGE [38] and the MIRCA2000 [39] crop calendars.

Combining the climate data and soybean growing season, we computed fourteen extreme climate indicators (Table 1) as predictors of soybean yield in history and future. The extreme climate indicators included extreme high and cold temperatures (cold degree days, CDD; extreme degree days, EDD; mean value of daily maximum temperature, Tmax; mean value of daily minimum temperature, Tmin; maximum value of daily maximum temperature, TXx; minimum value of daily minimum temperature, TNn), precipitation indicators (cumulative precipitation, Pr; maximum 5-days cumulative precipitation, Rx5day), drought (maximum continuous drought days, Dd; actual evapotranspiration from crop model minus Pr, Drought) and combined heat and drought disasters (mean value of daily mean temperature during Dd, Dt). Generally, the minimum and maximum limits of temperatures for soybean were 8 and 30 °C, D8 and D30, respectively [40–42]. In addition, soybean growing degree days (GDD) during soybean growing season were calculated. The trends of future extreme climate indicators were derived from the coefficients of a simply linear regression function for 10 years (10a).

**Table 1.** Extreme climate indicators for machine learning framework during soybean growing season.

| Extreme Climate Indicators | Descriptions | Unit |
| --- | --- | --- |
| CDD | Cold degree days, cumulative value of daily mean temperature < 8 °C | °C d |
| GDD | Growing degree days, cumulative value of 8 °C ≤ daily mean temperature ≤ 30 °C | °C d |
| EDD | Extreme degree days, cumulative value of daily mean temperature > 30 °C [42] | °C d |
| D30 | Days of daily maximum temperature > 30 °C [40] | d |
| D8 | Days of daily minimum temperature < 8 °C | d |
| Tmax | Mean value of daily maximum temperature | °C |
| Tmin | Mean value of daily minimum temperature | °C |
| TXx | Maximum value of daily maximum temperature | °C |
| TNn | Minimum value of daily minimum temperature | °C |
| Dd | Maximum continuous drought days (daily precipitation < 2 mm) [5] | d |
| Dt | Mean value of daily mean temperature during Dd [5] | °C |
| Pr | Cumulative precipitation | mm |
| Rx5day | Maximum 5-days cumulative precipitation (Vogel et al., 2021) | mm |
| Drought | The difference between actual evapotranspiration and Pr | mm |

### 2.3. Hybrid Model Framework

A hybrid model combining the crop model and extreme climate indicators with machine learning methods is proposed to constrain and reduce the uncertainty of soybean yield predicted by the crop model [15,43]. The process of the hybrid model is shown in Figure 1. At first, the GEPIC model is applied at the grid scale to obtain soybean yields, growing seasons and actual evapotranspiration. Subsequently, fourteen extreme climate indicators are derived from climate datasets during soybean growing seasons. Finally, soybean yield simulations from the GEPIC model and derived extreme climate indicators are applied as predictor variables in the hybrid model. After considering numerous machine learning algorithms, six main machine learning algorithms are analyzed to achieve the best performance of the hybrid model, which include MLR (Multi-Linear Regression), LASSO (the Least Absolute Shrinkage and Selection Operator), SVM (Support Vector Machine), RF (Random Forest), DT (Decision Tree) and KNN (K-Nearest Neighbor). The application of

machine learning algorithms is based on python3.7-Scikit-learn [44]. We use the grid-search algorithm of Scikit-learn to find the best parameters for the hybrid models. The total grid number is 123,604 × 5 climate datasets for training and 7090 × 5 climate datasets for validation.

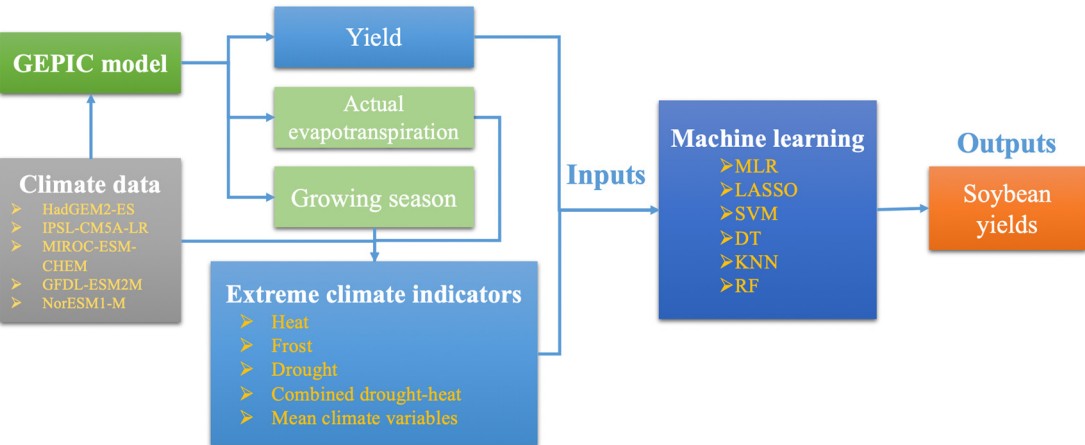

**Figure 1.** The process of the hybrid model.

The observed soybean yield is derived from the GDHY dataset where the yield data from 1981 to 2000 is used for training and that from 2001 to 2005 is for evaluation. The observed soybean yields are derived from the GDHY (the Global Dataset of Historical Yields for major crops) [45]. The GDHY dataset has combined statistical and satellite data with a spatial resolution of 0.5° × 0.5°.

Finally, the best hybrid model is used to project the future soybean yield. For each model simulation, long-term yield average and interannual yield variability are calculated to analyze the trends and uncertainties of the crop model and the hybrid model. Uncertainties are calculated as standard deviation for the soybean yield simulations from the GEPIC model and the hybrid models, similar to previous studies [15,43]. The predicted soybean yields are used to fit probability density functions (PDF) of kernel density estimate (KDE) functions. The interannual yield variability, uncertainties and KDE function were calculated as follows.

$$IYV = \frac{Y_{future} - Y_{baseline}}{Y_{baseline}} \tag{1}$$

where *IYV* is the interannual yield variability, $Y_{future}$ is future yield in one year or the future mean yield in a time period, $Y_{baseline}$ is the baseline mean yield.

$$U = \sqrt{\frac{\sum (Y - \overline{Y})^2}{n-1}} \tag{2}$$

where *U* is the uncertainties of a yield timeseries, $Y_i$ is yield, $\overline{Y}$ is the mean yield, *n* is the number of the yield timeseries.

$$f_h(x) = \frac{1}{nh} \sum_{i=1}^{n} K(\frac{Y - Y_i}{h}) \tag{3}$$

where $f_h(x)$ is the KDE function, *h* is a smoothing parameter which is estimated by python seaborn package, *K* is the Gaussian kernel, *Y* is any given yield point, $Y_i$ is a yield timeseries, *n* is the number of the yield timeseries.

### 2.4. Model Evaluation

Three consistency metrics are calculated to measure the agreement and disagreement between the original and improved simulated soybean yield. These evaluation metrics are

quantified by correlation coefficient (*R*), Root-Mean-Squared Difference (*RMSD*) recommended by Willmott [46], and Relative Root-Mean-Squared Difference (*rRMSD*).

$$R = \frac{\sum (S_i - \overline{S})(O_i - \overline{O})}{\sum (S_i - \overline{S})^2 \sum (O_i - \overline{O})^2} \tag{4}$$

$$RMSD = \sqrt{\frac{\sum_{i=1}^{n} (S_i - O_i)^2}{n}} \tag{5}$$

$$rRMSD = RMSD/\overline{O} \tag{6}$$

where $O_i$ is the observed data from GDHY dataset, $S_i$ is the simulated data from crop model or hybrid model, $\overline{O}$ is the mean value of observations, and $\overline{S}$ is the mean value of the simulations. The smaller the *RMSD*, the better the agreement between simulations and observations. The *R* provides the degree of association and ranges from −1 to 1. The *rRMSD* is derived from *RMSD* and gives a percentage measure of the relative differences between simulations and observations.

## 3. Results

### 3.1. Comparisons between GEPIC Model and the Hybrid Models during the Historic Period

Simulated soybean yield from the GEPIC model and the hybrid models have been evaluated at global scale (Figure 2). Before building the hybrid model, the GEPIC model has obviously poorest performance with the R, RMSD and rRMSD values of 0.138, 1.401 t/ha and 71.57%, respectively. The GEPIC+RF model performed best with the highest R value of 0.812 and lowest RMSD and rRMSD values of 0.716 and 36.62% respectively. The R values lower than 0.5 were those of the hybrid models GEPIC+MLR, GEPIC+LASSO and GEPIC+KNN. The rRMSD values higher than 50% were those of the hybrid models GEPIC+MLR, GEPIC+LASSO and GEPIC+SVM.

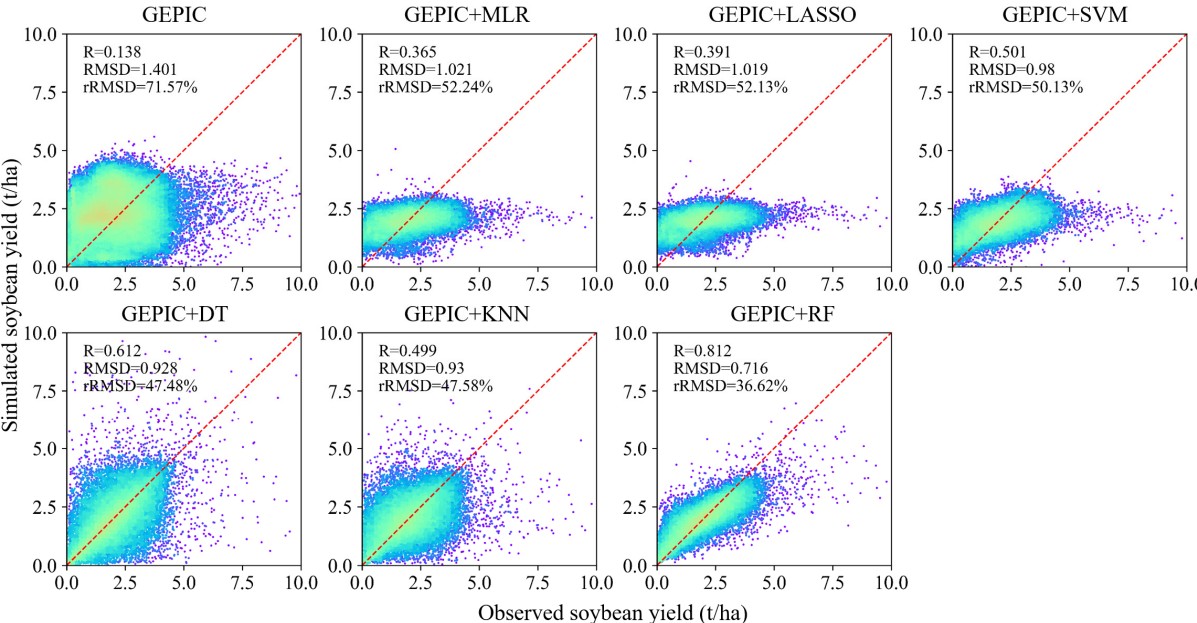

**Figure 2.** Simulated soybean yield comparisons between GEPIC model and six hybrid models during the historic period from 2001 to 2005 around the globe.

All the simulation evaluation metrics' values from the GEPIC model and the hybrid models at the national scale are tabulated in Table 2. We also estimated how the GEPIC and the hybrid models perform at the national and grid scale. Since the GEPIC+RF model performed best at the global and national scale, national simulations from the GEPIC+RF

model are displayed quantitatively in Figure 3 and the rRMSD values visually at grid scale in Figure 4. Similar to the comparisons at the global scale, the GEPIC+RF model at the national scale has the best performance with the closest 1:1 line and the highest R ranging from 0.513 to 0.733, as well as the lowest RMSD ranging from 0.716 to 1.132 t/ha and rRMSD ranging from 31.97% to 53.92%, for the five main producing countries. To be more specific, the GEPIC model performed best in China, while the lowest metrics were in Brazil of the five countries. Visually, when using the GEPIC+RF model, the central USA, Argentina, west India, Nigeria and Brazil have obvious decreasing RMSD values within 0.5–2 t/ha.

**Table 2.** Evaluation metrics of GEPIC and GEPIC+ML models for globe and main producing countries during the historic period.

|  |  | Argentina | Brazil | USA | China | India |
|---|---|---|---|---|---|---|
|  | GEPIC | 0.032 | 0.032 | 0.114 | 0.141 | 0.303 |
|  | GEPIC+RF | 0.569 | 0.513 | 0.629 | 0.579 | 0.733 |
|  | GEPIC+KNN | 0.305 | 0.308 | 0.42 | 0.266 | 0.338 |
| R | GEPIC+DT | 0.381 | 0.345 | 0.442 | 0.356 | 0.511 |
|  | GEPIC+MLR | 0.095 | 0.207 | 0.239 | 0.308 | 0.57 |
|  | GEPIC+LASSO | 0.063 | 0.21 | 0.286 | 0.355 | 0.579 |
|  | GEPIC+SVM | 0.395 | 0.293 | 0.504 | 0.41 | 0.456 |
|  | GEPIC | 1.362 | 1.584 | 1.423 | 1.008 | 1.355 |
|  | GEPIC+RF | 0.805 | 1.109 | 1.132 | 0.851 | 0.716 |
|  | GEPIC+KNN | 1.283 | 1.348 | 1.434 | 0.981 | 0.938 |
| RMSD | GEPIC+DT | 1.058 | 1.464 | 1.44 | 1.085 | 0.828 |
|  | GEPIC+MLR | 0.982 | 1.403 | 1.331 | 0.935 | 0.834 |
|  | GEPIC+LASSO | 0.965 | 1.375 | 1.367 | 0.933 | 0.812 |
|  | GEPIC+SVM | 0.996 | 1.501 | 1.414 | 0.917 | 0.824 |
|  | GEPIC | 54.07% | 58.93% | 52.22% | 51.00% | 102.00% |
|  | GEPIC+RF | 31.97% | 41.24% | 41.56% | 43.06% | 53.92% |
|  | GEPIC+KNN | 50.96% | 50.14% | 52.65% | 49.61% | 70.60% |
| rRMSD | GEPIC+DT | 42.00% | 54.47% | 52.86% | 54.86% | 62.31% |
|  | GEPIC+MLR | 38.98% | 52.21% | 48.85% | 47.31% | 62.74% |
|  | GEPIC+LASSO | 38.30% | 51.15% | 50.17% | 47.16% | 61.11% |
|  | GEPIC+SVM | 39.55% | 55.83% | 51.89% | 46.37% | 62.01% |

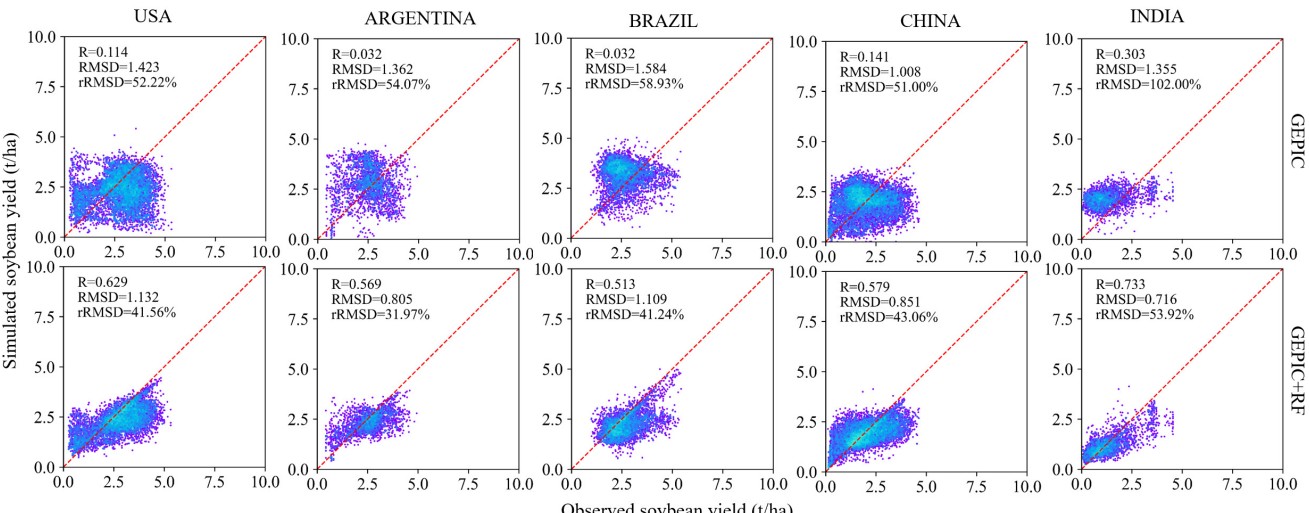

**Figure 3.** Simulated soybean yield comparisons between GEPIC model and GEPIC+RF during the historic period from 2001 to 2005 for the main producing countries.

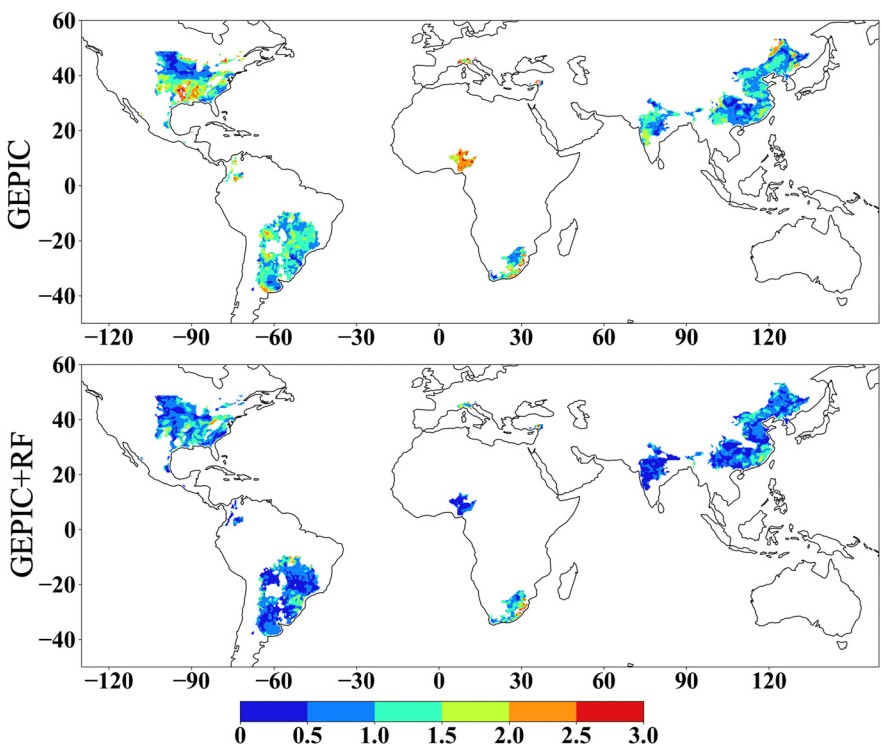

**Figure 4.** Spatial distribution comparisons of RMSD (t/ha) between the GEPIC and GEPIC+RF model.

### 3.2. Importance Analysis of Extreme Climate Indicators for Soybean Yield

We used the RF model to estimate the importance (number of y axis) of extreme climate indicators for the globe and the five main soybean producing countries (Figure 5). The higher the importance value is, the more decisive the extreme climate indicator was. At the global scale, the most affecting extreme climate indicator was the Tmin, followed by Rx5d, GDD and TNn. The most affecting extreme climate indicator was drought for China and the USA, and D8 for Argentina. Combined heat and drought disaster (Dt) dominated for India's soybean yield, followed by drought. For Brazil, the TNn and Drought ranked first and second, respectively. We can infer that for the northern hemisphere, drought dominates for soybean yield and low temperature disaster is the main affecting factor for the southern hemisphere. The values of the importance would help us to assess the priority of an extreme climate indicator and improve adaptation and mitigation ability.

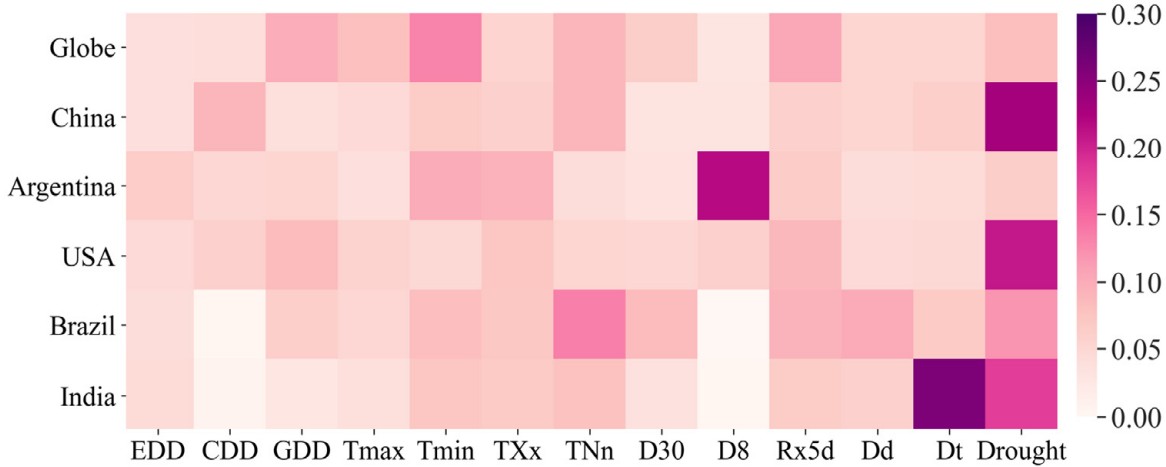

**Figure 5.** Heat map of extreme climate indicators' importance for soybean yield.

### 3.3. Future Projections of Soybean Yield

For the future period of 2040 to 2099, the projected annual mean soybean yields from the GEPIC and GEPIC+RF models are shown in Figure 6. Both the GEPIC model and GEPIC+RF model witness a slight increase in the future, with the positive slopes and the coefficients of determination ($R^2$) being very small. There were obvious overestimations occuring at the global scale for the GEPIC model, and there are larger fluctuations of soybean yield compared to the GEPIC+RF model. Specifically, the mean differences between the two models are 0.42 t/ha for global, 0.54 t/ha for the USA, 0.48 t/ha for China, 0.40 t/ha for Argentina and India, and 0.28 t/ha for Brazil. Brazil has the lowest impacts from climate disasters, while the USA is affected most by climate disasters.

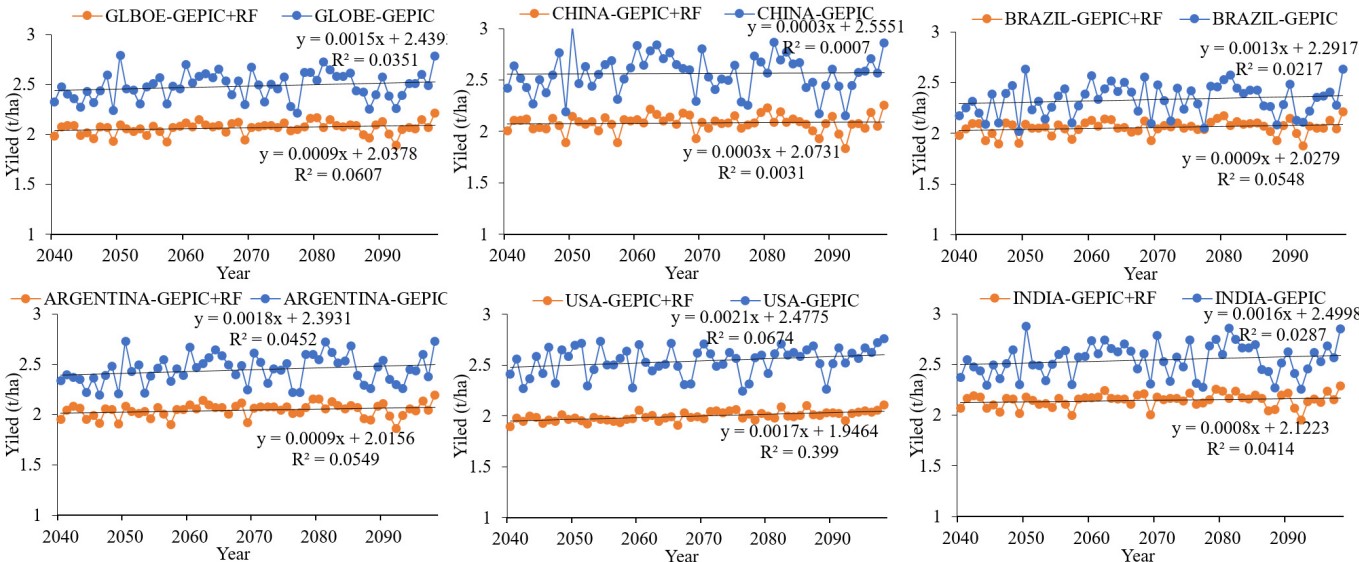

**Figure 6.** Future projections of soybean yield from the GEPIC model and GEPIC+RF model for 2040 to 2099. The black lines are linear trend line.

For the three 20-year periods of 2040–2059, 2060–2079 and 2080–2099, the GEPIC model projects larger variations and higher yields than the GEPIC+RF model for the whole globe and for the five main producing countries (Figure 7). The GEPIC model overestimated soybean yields compared with the baseline yield (mean yield of 1981–2005), and there are small differences in the mean yield variations between the three 20-year periods. The GEPIC+RF model shows the variations range from −10.89% to 4.39% for the globe compared with the baseline yield, while the variations of the GEPIC model range from 5.53% to 31.75%. For the main soybean producing countries, the GEPIC model projects that India has the largest increase, with a yield variation of 102.68%, followed by China with 45.01%, while there is a reducing tendency with −4.13% in the USA compared with baseline. On the contrary, the GEPIC+RF model projects that the yield variations are 74.74%, 17.71%, −24.66%, −12.55% and −14.43% for India, China, the USA, Argentina and Brazil, respectively.

Future soybean yield change patterns from 2040 to 2099 are displayed in Figure 8, which shows an increase of more than 0.05 t/ha/10a for eastern Brazil, most regions of China, Nigeria, northern India and the central and northern USA based on the GEPIC model. On the other hand, the GEPIC+RF model shows that the soybean yields will increase more than 0.05 t/ha/10a only in northern and northeast China, Nigeria and northern Argentina. The increasing trends show significant reduction in eastern Brazil, the central USA and southern China.

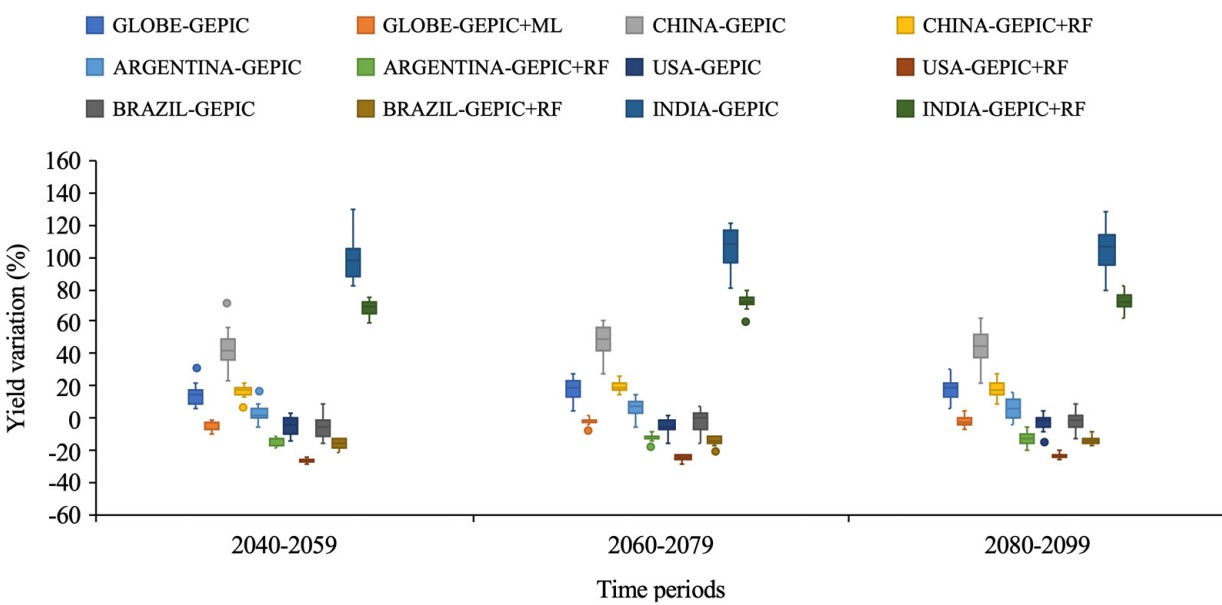

**Figure 7.** Boxplot of soybean yield variations for 20-year periods in the future compared with the baseline (1981–2005) mean yield.

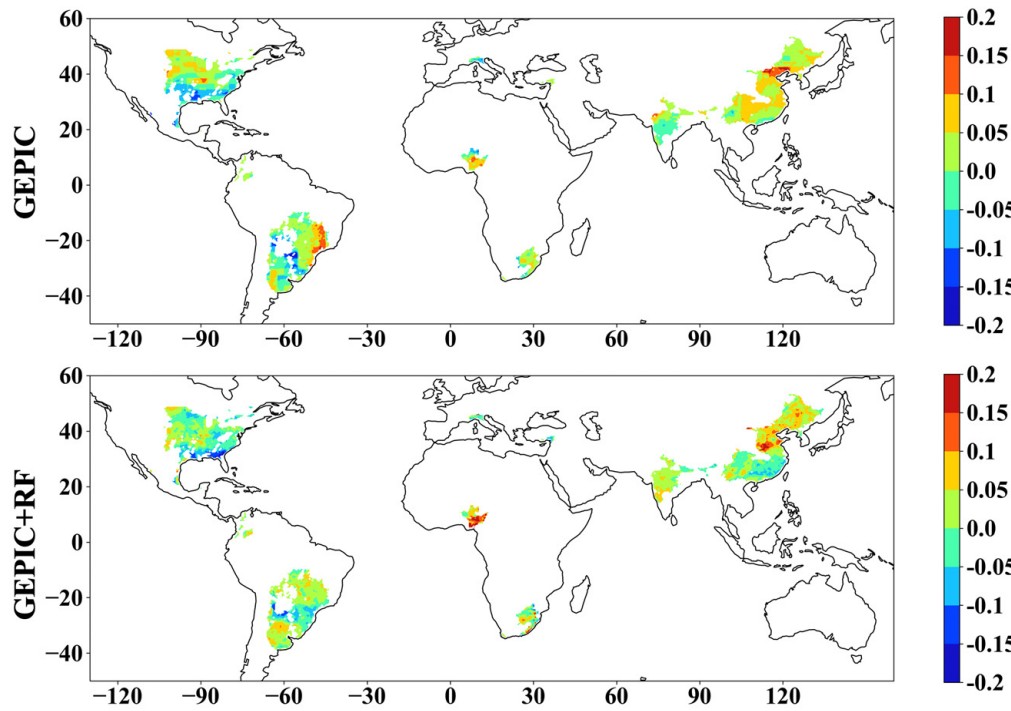

**Figure 8.** Soybean yield change trends from 2040 to 2099 (t/ha/10a).

From the importance analysis results (Figure 5), the changes of main extreme climate indicators during soybean growing season in the future are analyzed (Figure 9). The Tmin has an increasing trend at global scale and has a more than 0.3 °C/10a trend in the central-south USA, northern Argentina and southern Brazil. The D8 has a decreasing trend in high latitude areas in the central-north USA, northeastern China and northern Argentina. Drought has an increasing trend in the central-south and eastern USA, southern Brazil, northern Argentina, Nigeria, and southwestern China, while it has a significant decreasing trend in central India and Brazil. The combined drought–heat disaster, Dt, tends to increase in most regions in all of the main producing countries. Moreover, the values of Dt have an obvious increase trend in southwestern and central China, the central USA,

northern Argentina and southern Brazil where Dt is higher than 0.5 °C/10a. The TNn has a significant increase trend higher than 0.5 °C/10a in the central-south USA, central Brazil, and southwestern and southern China.

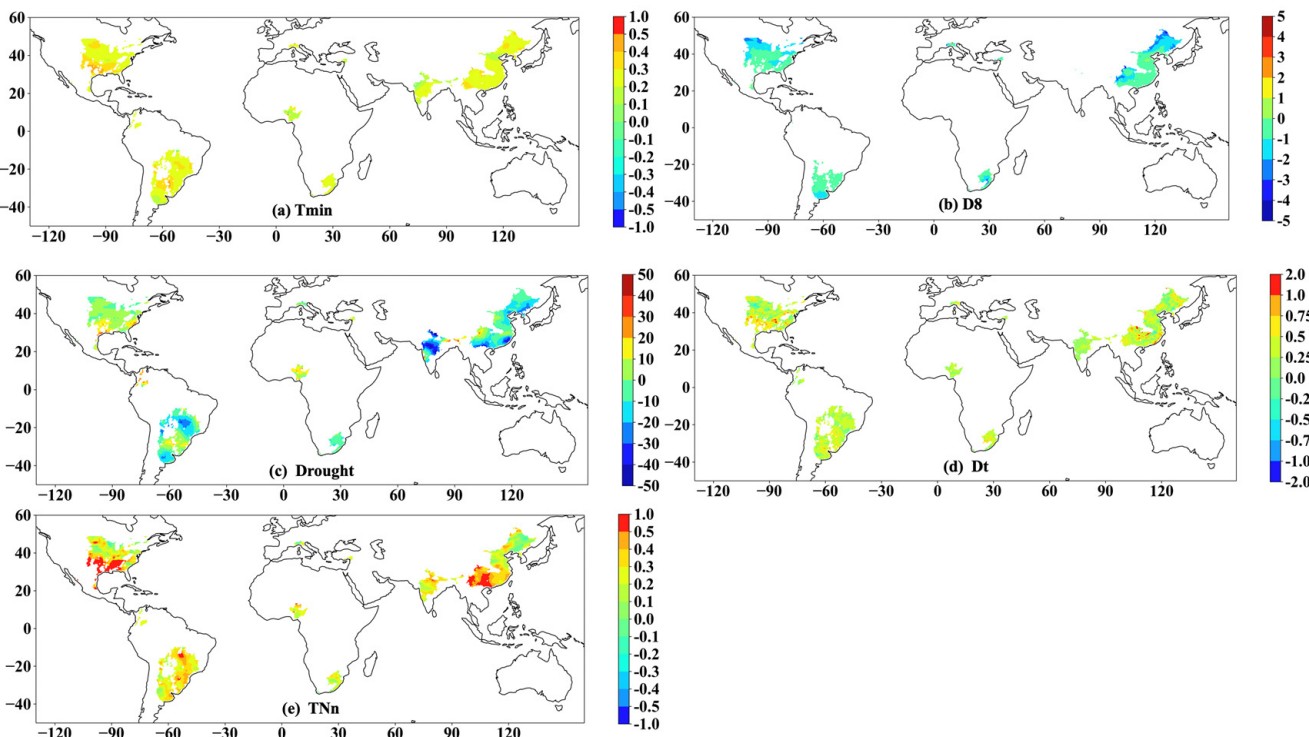

**Figure 9.** The trends of extreme climate indicators during soybean growing season from 2040 to 2099 (per 10a). (**a**) Tmin (°C/10a) stands for mean value of daily minimum temperature. (**b**) D8 (days/10a) stands for days of daily minimum temperature < 8 °C. (**c**) Drought (mm/10a) stands for the difference between actual evapotranspiration and cumulative precipitation. (**d**) Dt (°C/10a) stands for mean value of daily mean temperature during maximum continuous drought days (daily precipitation < 2 mm). (**e**) TNn (°C/10a) stands for minimum value of daily minimum temperature.

### 3.4. Uncertainties in Future Yield Projection

The PDF of future changes in soybean yield compared to the baseline mean yield (1981–2005) from the GEPIC model (unconstrained) and GEPIC+RF model (constrained) for the globe and main producing countries are shown in Figure 10. An obvious uncertainties reduction is achieved not only for the globe, but also in the five main soybean producing countries. A large uncertainty of 127.70% is found for the globe from the GEPIC model. After being constrained by machine learning method, the uncertainty decreases to 87.16%. The uncertainties have been reduced by about 41.83%, 31.83%, 31.88% and 28.45% for China, Argentina, the USA, Brazil and India, respectively. The largest uncertainties of the GEPIC+RF model are for the USA, while the smallest uncertainties are for India. The constrained results project that there are more soybean yield losses in the future, and this indicates that crop models may overestimate the soybean yield.

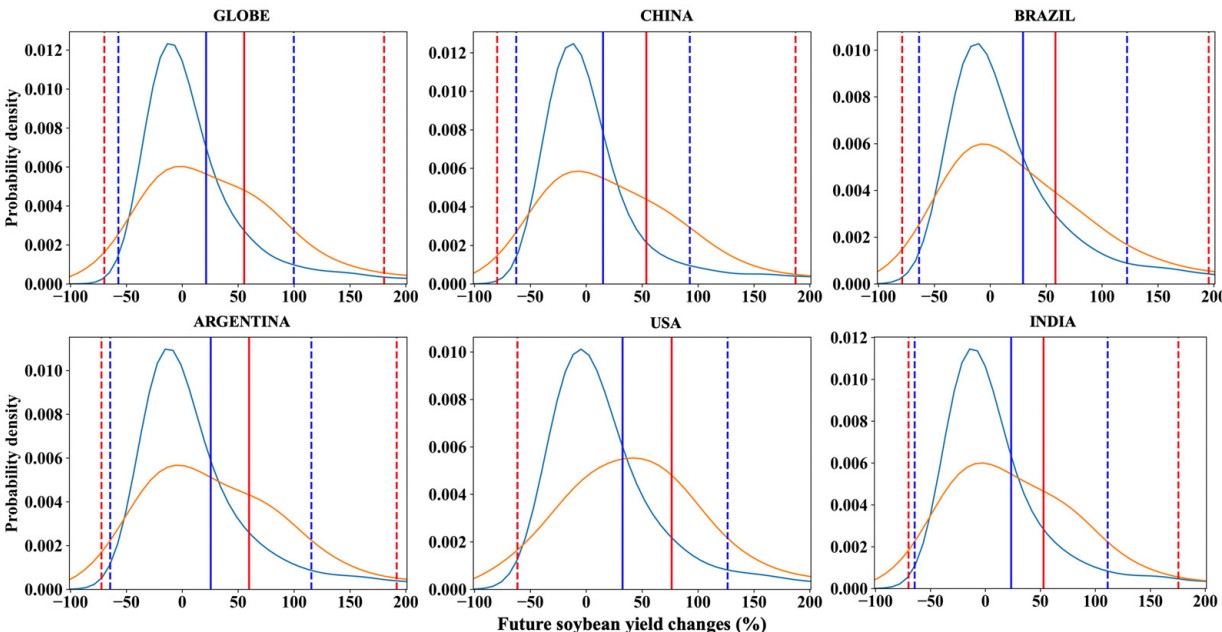

**Figure 10.** Probability distribution of future (2040–2099) soybean yield uncertainties compared with mean yield of baseline (1981–2005) for the globe and main producing countries. The blue and orange line are the probability distribution of the simulated soybean yield from the GEPIC and hybrid models, respectively. The vertical solid and dash lines indicate the mean and ± standard deviation of future soybean yield changes.

## 4. Discussion

### 4.1. Performance of the Hybrid Models

Recent studies have found that there were some overestimations and large uncertainties from the process-based crop model [13,21]. In this study, we proposed a hybrid model by combining multiple machine learning algorithms and the GEPIC model to improve the simulated yield given by the GEPIC model. The GEPIC+RF model had the best simulation ability to reconstruct historical observed soybean yield compared to MLR, SVM, LASSO, DT and KNN models (Figure 2). Compared with the original GEPIC model, the R value of the GEPIC+RF hybrid model during the historical period has increased by 370%, and RMSD and rRMSD values decreased by 36%. In the future, when applying the hybrid model, the uncertainties and overestimations of simulated soybean yield would reduce by 28.45–41.83% and 13.40–27.23%, respectively. The desirable performance of the GEPIC+RF hybrid model is probably related to the RF model being able to solve the overfitting in DT and better explore complex non-linear relations with multiple trees in agricultural-based applications [15,24]. Moreover, coupling the crop model and extreme climate indicators is an effective and robust way to project crop yield and can be easily expanded to the impacts of extreme climate on other crops.

### 4.2. Importance and Changes of Climate Indicators

Extreme climate events, as the main factor causing yield losses, are attracting more and more concern recently [9,14,47]. Since there are complex non-linear relationships between climate indices and soybean yield, it is crucial to identify the main extreme climate drivers globally and for the main producing countries. We found that low-temperature-related indices have a decreasing trend while high-temperature-related indices have a reversed trend during the soybean growing season in the future. The low-temperature-related indices are the main affecting feature for crop yield for the globe and for the southern hemisphere countries of Brazil and Argentina. However, drought and drought–heat disasters are the main affecting features for the northern hemisphere countries of China, the USA and India, which is consistent with Zipper et al. [16] and Leng et al. [3].

*4.3. Limitations*

However, there are some limitations in this study. The uncertainties of the simulated yields are possibly from the observed gridded crop yield dataset, climate datasets, and the process-based crop model and machine learning algorithms [48]. Even though high-resolution observed-yield datasets are hard to access now, international collaboration can obtain high-resolution observed-yield data. Moreover, with the improvement of climate models and multi-model ensemble methods, the accuracy of future climate projections are improving based on the Coupled Model Intercomparison Project Phase 6 [49]. For the process-based crop model, in future study we should enhance the scientific understanding of the process of extreme climate events tolerance for crop growth and yield [22,23], or use a multi-crop model ensemble to reduce uncertainties [35]. Although the uncertainties have been greatly reduced, machine learning still has its own uncertainties which affect the statistical model and crop model. Moreover, quantifying the uncertainties from each model and the hybrid model is still unknown and needs further research [50]. In addition, this study ignored technology development and agronomic adjustments in the future. The agricultural technology development [51] and agronomic adjustments [52] may still not completely offset future climate trends and extreme climate, and they will also increase the uncertainty of yield prediction [53].

At the same time, our study applied mean climate state indices and extreme climate indices as input feature variables in the hybrid model, but more climate indices such as monthly mean climate variables [14] and VPD [42] could be tested as ML inputs. Furthermore, the variables selection for machine learning is important, but manual feature engineering is tedious and time consuming [54]. In future research, deep learning models with more hidden layers will be capable of learning feature representations from data in an end-to-end regime instead of using manual feature engineering based on human experience and prior knowledge [55].

Finally, to cope with the future climate extremes for soybean, appropriate adaptation and mitigation strategies should be developed [47,56]. For example, adding more fertilizer [57], controlling the soybean growing window [58] and planting longer-duration soybean cultivars [59] are efficient ways to mitigate negative climate impacts in the future. Later planting dates and drought-tolerant soybean cultivars are more suitable for the USA [60], China [61], and South America [62–64] in the future. Meanwhile, early warning systems and contingency planning are very important for developing countries [64].

**5. Conclusions**

Since soybean yield is susceptible to extreme climate, in this study we constructed hybrid models by coupling the GEPIC model and extreme climate indicators with six machine learning models which were then compared to present a robust hybrid model to explore the impacts of extreme climate events and reduce the yield prediction uncertainties for current and future soybean yields. It was found that the process-based crop model has larger uncertainties, and would overestimate soybean yield and underestimate risks from extreme climate events, but these problems can be eliminated with machine learning methods. This study demonstrates that the hybrid model of the GEPIC+RF model has the best performance compared with the GEPIC model and the other five GEPIC + ML models, both at the global and country scales. Tmin is the main affecting weather disaster factor for global soybean yield, while drought, Dt and D8 are the main factors for China and the USA, India and Argentina, respectively. Future soybean yield projections from the process-based crop model may have a 13.40–27.23% overestimation, 0.42 t/ha on average, at the global scale due to the ignorance of extreme climate events. With the optimal GEPIC+RF model, the uncertainties are reduced by 28.45–41.83% for future soybean yield projections both at global and country scales. Our results imply that the extreme climate impact functions in the process-based crop model need further improvement and the hybrid model can serve as a powerful tool for crop yield simulation. This study can provide some useful information for global soybean traders, farmers and policy makers with regard to agricul-

tural adaptation and mitigation strategies in the context of increased climate extremes in the future.

**Author Contributions:** Conceptualization, Q.S. and X.C.; methodology, Q.S., Y.Z. and X.C.; validation, S.C.; writing—original draft preparation, Q.S., Y.Z. and X.C., writing—review and editing, Q.S., X.C., Q.Y., X.Z. and A.F.; data curation, S.C., X.Z. and A.F. All authors have read and agreed to the published version of the manuscript.

**Funding:** This research was funded by the Basic Scientific Research Operating Expenses of the Chinese Academy of Surveying and Mapping (AR2205), the National Natural Science Foundation of China program (Grant No. 41901379), the National Key Research and Development Program of China (Grant No. 2019YFD1002200), and the Basic Research Fund of CAMS (Grant No. 2020Y003 and 2021Z010).

**Institutional Review Board Statement:** Not applicable.

**Informed Consent Statement:** Not applicable.

**Data Availability Statement:** The datasets generated during and/or analyzed during the current study are available from the corresponding author on reasonable request.

**Acknowledgments:** For their roles in producing, coordinating, and making available the ISIMIP input data and impact model output, we acknowledge the modelling groups, the ISIMIP sector coordinators and the ISIMIP cross-sectoral science team.

**Conflicts of Interest:** The authors declare no conflict of interest.

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
