# Peer review of "Coupling Process-Based Crop Model and Extreme Climate Indicators with Machine Learning Can Improve the Predictions and Reduce Uncertainties of Global Soybean Yields"

_agriculture, doi:10.3390/agriculture12111791_

Round 1

Reviewer 1 Report

The article “Coupling process-based crop model and extreme climate indicators with machine learning can improve the predictions and 3 reduce uncertainties of soybean yields” proposed a hybrid model to constrain such uncertainties by coupling GEPIC, and GEPIC+RF among other models for extreme climate indicators. Overall, the paper is alright however, English language, and grammar should be improved in different sections of the paper. Abstract should be improved from line 18-23.

1. Introduction: Minor updates, language and improve references in number.

2. Materials and methods: This should be improved by providing the idea of how the data was collected. I completely understand that the data was taken from an application, and some might have conducted the field experiments too simultaneously, however, this is what you need to write in a detailed form. The one that is provided at this time to the readers is not optimum. Very hard to understand. 

3. Line 97: The mentioned parameters, fourteen extreme Climate Indicators [such as Cold degree days, cumulative value of daily mean temperature, Days of daily maximum temperature, Days of daily minimum temperature, mean value of daily maximum temperature, Mean value of daily minimum temperature and so on....] are not explained! In my opinion, they are the climate indicators, if it variates in maximum or minimum then they could be, but you need to explain how they are used as the extreme Climate Indicators?

4. Line 143-150: Not clear. Equation not explained, please explain it at the following text.

5. Figure 2: Enlarge the numbers of the figure and explain it I a better way in line 172-180.

6. Figure 3, 4, 5, 6, 7, 8, 9: Same as above.

7. Conclusion: Need to improve, no bullet points, update according to the objectives of the paper. At this moment, this section needs further improvement and have a huge gap in its innovation.

Thanks.

Reviewer 2 Report

Please, find the comments and suggestions in ralation to manuscript with ID agriculture-1969046 attached.

Round 2

Reviewer 1 Report

Many things are improved, however, as mentioned previously:

Line 143-150: Not clear. Equation not explained, please explain it at the following text.

5. Figure 2: Enlarge the numbers of the figure and explain it I a better way in line 172-180.

6. Figure 3, 4, 5, 6, 7, 8, 9: Same as above.

Reviewer 2 Report

The authors made significant changes to their manuscript. I thank you for your replies and justifications and accept them.

Author Response

Thank you a lot for your helpful suggestions again.